# Neural Activity for Uninvolved Knee Motor Control After ACL Reconstruction Differs from Healthy Controls

**DOI:** 10.3390/brainsci15020109

**Published:** 2025-01-23

**Authors:** Meredith Chaput, Cody R. Criss, James A. Onate, Janet E. Simon, Dustin R. Grooms

**Affiliations:** 1Division of Physical Therapy, School of Kinesiology and Rehabilitation Sciences, College of Health Professions and Sciences, University of Central Florida, Orlando, FL 32816, USA; meredith.chaput@ucf.edu; 2Department of Radiology, Medical University of South Carolina, Charleston, SC 29425, USA; cross@musc.edu; 3Division of Athletic Training, School of Health and Rehabilitation Sciences, College of Medicine, The Ohio State University, Columbus, OH 43210, USA; onate.2@osu.edu; 4Ohio Musculoskeletal and Neurological Institute, Ohio University, Athens, OH 45701, USA; simonj1@ohio.edu; 5Department of Athletic Training, School of Applied Health Sciences and Wellness, College of Health Sciences and Professions, Ohio University, Athens, OH 45701, USA; 6Department of Physical Therapy, School of Rehabilitation & Communication Sciences, College of Health Sciences and Professions, Ohio University, Athens, OH 45701, USA

**Keywords:** fMRI, cognition, neurocognition, knee, motor control

## Abstract

Recovery from anterior cruciate ligament reconstruction (ACLR) induces bilateral functional and physiological adaptations. Neurophysiologic measures of motor control have focused on the involved knee joint, limiting understanding regarding the extent of bilateral neural adaptations. Therefore, the aim of this study was to investigate differences in neural activity during uninvolved-limb motor control after ACLR compared to healthy controls. Methods: Fifteen participants with left ACLR (8 female and 7 male, 21.53 ± 2.7 years, 173.22 ± 10.0 cm, 72.15 ± 16.1 kg, Tegner 7.40 ± 1.1, 43.33 ± 33.1 mo. post-surgery, 2 patellar tendon, and 13 hamstring) and 15 matched controls (8 female, 23.33 ± 2.7 years, 174.92 ± 9.7 cm, 72.14 ± 15.4 kg, Tegner 7.33 ± 1.0) participated. Neural activity was evaluated using functional magnetic resonance imaging on a 3T Siemens Magnetom scanner during four 30-s cycles of a right (uninvolved) knee flexion-extension task paced with a metronome (1.2 Hz) and was completed interspersed with 30 s of rest. A significance threshold of *p* < 0.05 was used for all analyses, cluster corrected for multiple comparisons, and z-thresholds of >3.1 (subject level), and >2.3 (group level). Results: The ACLR group had greater neural activity in one statistically significant cluster corresponding to the left middle frontal gyrus (MFG) (834 voxels, z = 3.81, *p* < 0.01 multiple comparisons corrected) compared to controls. Conclusions: These data indicate a potential contribution to uninvolved-knee neuromuscular deficits after injury and support the limitations of using the uninvolved side as a clinical reference. Uninvolved knee motor control after ACLR may require greater cognitive demand. Clinicians should be aware that the uninvolved limb might also demonstrate whole brain alterations limiting clinical inference from functional symmetry.

## 1. Introduction

Anterior cruciate ligament (ACL) rupture is a common sports related knee injury occurring frequently through non-contact mechanisms. The non-contact injury mechanism is in part secondary to a sensorimotor error resulting in poor neuromuscular control and knee valgus collapse [1,2]. Reconstructive surgery (ACLR) and rehabilitation are the mainstay treatments for restoring mechanical joint stability but do not guarantee normalization of knee sensorimotor control which contributes to a large number of athletes being unable to return to their prior level of competition [3]. Despite surgery and rehabilitation, persistent knee sensorimotor impairments have been documented including decreased quadriceps strength and central activation ratio [4], altered coordination during gait [5], and deficits in functional performance [6]. The inability to holistically restore sensorimotor control likely contributes to the one in four ACL injury rate in athletes attempting return to high level activity. However, half of second ACL injuries occur to the contralateral (uninjured) extremity [7], indicating there may be central nervous system (CNS) adaptations contributing to reinjury risk not only to the involved limb but the uninvolved side as well.

Contralateral limb ACL injury-risk may be due, in part, to CNS adaptations that occur in consequence to the original injury and/or surgical intervention such as pain, inflammation, joint injury, and ligamentous deafferentation [8,9]. Konishi [10] has demonstrated that after ACLR, gamma loop dysfunction exists bilaterally throughout recovery and results in quadriceps activation attenuation [10]. Further, Zarzycki et al. [11] demonstrated early after ACLR, the resting motor threshold of the primary motor cortex, responsible for uninvolved limb quadriceps activation was greater relative to healthy controls [11]. A higher resting motor threshold (e.g., reduced corticospinal excitability) indicates that a greater level of neural activation is necessary to transmit a descending efferent potential within the corticospinal tract, providing some etiological evidence as to why bilateral quadriceps muscle strength and functional deficits are observed after ACLR [12,13].

To date, studies investigating whole brain alterations after an ACL injury have been primarily limited to involved limb movement during simple knee flexion-extension [14], heel-slide [15], force matching [16], and proprioceptive [17] type paradigms. Collectively, these studies demonstrate widespread alterations in the CNS for sensorimotor control of knee movement with higher levels of neural activity occurring in brain regions responsible for cognitive processing (prefrontal cortex, precuneus, and posterior cingulate cortex), visual-spatial memory (precuneus and intracalcarine cortex), cross-modal sensory integration (lingual gyrus), and descending motor control (corticospinal tract) [15,18,19,20]. Additionally, as early as six-weeks after ACLR, single limb stance on the injured limb requires increased functional connectivity in the fronto-parietal, fronto-occiptial, and occipito-parietal regions [21]. In the late stages of recovery, individuals with ACLR exhibit lower sensorimotor and higher motor planning demands with higher levels of motor inhibition compared to healthy controls for injured limb balance [22]. Thus, research continues to expand on the bilateral whole brain cortical changes secondary to injury. However, despite ACL injury being a unilateral ligamentous disruption (i.e., deafferentation event), CNS adaptations are not limited to only the involved knee. Bilateral spinal and supraspinal alterations in quadriceps neuromuscular responses occur [4,10,11,13], however, there is a paucity of evidence that has investigated whole brain adaptations for uninvolved knee movement in those with a history of ACLR.

Therefore, the current study aimed to fill this gap by quantifying whole-brain neural activity during uninvolved (healthy limb) movement, using functional Magnetic Resonance Imaging (fMRI) after ACLR compared to matched uninjured controls. Since neurophysiological dysfunction presents bilaterally after ACLR, we hypothesized that the ACLR group would demonstrate alterations in neural activity during uninvolved knee movement in sensorimotor and/or cognitive regions relative to healthy matched controls.

## 2. Materials and Methods

### 2.1. Participants

This study was approved by the Ohio State University Institutional Review Board (#2012H0273) and informed consent was obtained prior to study enrollment. This study enrolled 15 individuals with a history of left unilateral ACLR and 15 healthy matched controls aged 18 to 35 years (Table 1). Individuals were matched on age, sex, bodyweight, and activity level (Tegner > 6) and all were actively engaged in a bachelor’s level education or had graduated from a four-year university. ACLR participants were at least 6 months to 5 years from surgery and had received clearance for unrestricted physical activity from their surgeon, and control participants had no history of lower extremity injury. ACLR subjects with history of a contralateral injury, revision surgery, or any participant with a concussive history in previous 12 months were excluded from the study. Both groups demonstrated MRI compliance (e.g., no ferrous metal implants, claustrophobia, etc.). The healthy control group demonstrated similar demographics to the ACLR group (age, height, weight, and activity level), no history of lower extremity injury requiring surgery, and were compliant with MRI safety precautions.

### 2.2. MRI Paradigm and Data Acquisition

Preceding the fMRI data collection, participants first practiced the motor task. They lay supine on a treatment table with a bolster under their legs to provide both leg and trunk support. Bilateral ankles were also immobilized with a dorsiflexion night splint and arms were placed by their sides. Participants viewed a black computer screen to mimic the visual display they would see within the MRI. Participants began with 30 s of rest until a standardized auditory stimulus instructed them to “prepare for the next exercise, contract” and a 1.2 Hz metronome would begin. Participants then paced their knee flexion/extension to the metronome for 30 s until they were instructed by the audio file to “relax”. This was repeated for 5 rest and 4 movement blocks. Participants were instructed to minimize head motion as much as possible and to focus on matching the flexion/extension of their “kick” to the beat of the metronome. The research staff manually monitored the participants’ ability to keep pace with the auditory cue and provided both manual and verbal instruction as necessary during the training session. Participants practiced the task for at least 1 complete run, but as much as necessary to ensure neural activity measured in the MRI was not due to task novelty.

fMRI data were collected on a 3.0-T MAGNETOM (Siemens AG, Munich, Germany) scanner using a 12-channel array, receiver-only head coil. Participants lay supine on the MRI table with a wedge under their legs, ankle dorsiflexion splint to restrict ankle motion during the task, and protective earphones. Participants also were restricted in motion with a strap placed across their pelvis and one at the chest. Within the MRI head coil, MRI safe head padding was comfortably placed. Each session began with a high resolution structural T1-weighted image followed by a lower resolution functional scan during a motor task. During the functional scan, blood-oxygen-level dependent (BOLD) signal, an adjunct for neural activity [23,24], was measured during four-blocks of 30 s right (uninvolved) knee flexion/extension (beginning at 45° of flexion to terminal knee extension [Figure 1]) which was interweaved with five-blocks of 30 s rest [14]. The knee flexion/extension tasks was temporally cued with a metronome paced at 1.2 Hz during each 30 s movement block and research staff monitored repetitions to ensure consistent performance [25]. Each functional scan consisted of 90 whole-brain gradient-echo, echoplaner scans: TR = 3000 ms; TE = 28 ms; field of view = 220 mm; flip angle = 78°; slice thickness = 2.5 mm; voxel size = 2.5 mm^3^ for 55 slices [14].

Preprocessing of subject level fMRI data was completed using FSL (FMRIB, Oxford, UK, Version 6.0.7.15) software and consisted of brain extraction, MCFLIRT motion correction, Gaussian kernel FWHM 6 mm spatial smoothing, and mean-based BOLD intensity normalization of all volumes [26,27,28]. Automatic Removal of Motion Artifacts was used to reduce any motion-induced signal. After denoising, fMRI data were preprocessed using a high pass filter at 90 Hz. All functional (task-based) scans were co-registered with structural T1-weighted images in FSL using non-linear image registration to the standard Montreal Neurological Institute template 152.

The subject level analysis was the contrast of knee flexion/extension movement to rest and the group level analysis consisted of the paired contrast between individuals with ACLR and matched controls. Z statistic images were constructed nonparametrically with a significance threshold of *p* < 0.05 from Gaussian Random Field Theory cluster corrected for multiple comparisons and maximum z-score thresholds of >3.1 (subject level) and >2.3 (group level) [29,30]. The general linear model was used for both subject and group level analyses to examine movement relative to rest contrasts and a pairwise analysis of whole-brain group-average activation maps, respectively. Additionally, to determine if there were differences between groups in age, height, weight, and activity level, separate independent samples t-tests were used with the alpha set to 0.05 (Table 1).

## 3. Results

There were no significant differences between ACLR and control groups among demographic variables (*p* > 0.05, Table 1). The respective-group-average-activation patterns are shown in Figure 2. Figure 3represents the group level contrast between ACLR participants and the controls. ACLR participants demonstrated greater neural activity within a single cluster of the left middle frontal gyrus/frontal pole (MFG) (834 voxels, z = 3.81, *p* < 0.001, [Table 2]) compared to the controls for uninvolved limb movement. No other differences were identified between groups (Figure 3).

## 4. Discussion

The current study aimed to preliminarily investigate differences in neural activity associated with uninvolved limb movement following ACLR relative to matched controls. In support of our hypothesis, ACLR participants demonstrated different neural activity relative to controls, with increased activation occurring within a single cluster of the MFG.

After ACLR, the uninvolved limb has been considered an acceptable reference standard (i.e., limb symmetry index) for clinical assessments of muscle strength and functional performance. Limb symmetry for both quadriceps strength and functional performance has been shown to overestimate knee function and also relate to second injury risk [31]. However, after injury neurophysiological alterations such as impaired early rate of torque development, a metric that reflects motor unit recruitment/discharge rate, and central activation ratio failure, a metric quantifying the number of motor units recruited during maximal voluntary contraction, occur bilaterally [32]. Additionally, after ACL injury, one in three individuals experience central activation failure of their uninvolved limb quadriceps muscle [32], activating 12–19% less of the available motor units relative to healthy controls [33,34]. Additional mechanisms that explain prolonged contralateral quadriceps activation failure after ACLR could stem from altered brain activation, as those with ACLR require greater frontal cortex activity to complete the same quadriceps-dominant motor task as healthy controls.

To our knowledge, only one other study has evaluated uninvolved-knee neural activity for knee movement using fMRI. Schnittjer et al. [19], contrasted the neural activity between limbs (Uninvolved > Involved) and then compared this activity to uninjured persons. The authors found that for the uninvolved limb of ACLR individuals, the peak activation coordinate in a cluster within the frontal cortex was shifted anteriorly by 6.24 mm and inferiorly 7.7 mm relative to controls. Additionally, a second cluster within the parietal cortex demonstrated peak activation 7.25 mm medially and 12.08 mm interiorly compared to controls. This relative shift in anterior and inferior peak-activation location for the uninvolved after ACLR within the frontal cortex corresponds to our findings of increased MFG activity (frontal cortex) compared to controls.

Although limited research has evaluated neural activity for uninvolved-limb knee control, a study by Lepley et al. [35] found greater frontal cortex activity while performing a knee flexion-extension task on the involved limb in ACLR participants [35].These results are complemented by Baumeister et al. [16,17] who found greater frontal cortex activity measured by electroencephalography (EEG) during a force reproduction task [16] and joint reposition sense [17] assessment also of the involved extremity. The authors of these studies conclude that the heightened frontal cortex activity is likely secondary to increased neurocognitive demand and/or processing required after ACLR to perform knee motor control tasks. The current study expands on prior findings of greater frontal cortex activity and alterations in peak voxel location, indicating a potential similar neurologic deviation to control the “healthy” contralateral limb. Without the direct sensory perturbation of the lost ligament, the increased neural activity for contralateral knee movement is likely secondary to different mechanisms, perhaps due to attentional or motor planning compensations.

In a broad sense, the prefrontal cortex is associated with cognitive control functions such as sustaining attention, inhibiting habitual responses, and navigating dual-task scenarios [36]. The MFG plays a role in movement preparation, activating from 200 to 900 milliseconds prior to the onset of detectible goal-directed movement [37]. Additionally, the MFG has been identified as an active cortical region during a temporally paced reproduction button pressing task [38]. When temporally cued (cognitively attending to the time duration needed to complete a motor act) participants demonstrated greater MFG activity as compared to when no temporal cue was present [38]. These studies indicate the MFG both plans and assists in the cognitive execution of goal-directed behavior. Our study paradigm did not assess neural activity prior to movement, onset; however, it did include a temporally cued (metronome) knee flexion-extension task. Therefore, elevated activity within the MFG in the ACLR group but not controls might suggest greater cognitive attention or reduced efficiency for either the motor planning or the execution of precisely timed lower limb movement. Future work may consider dual-task protocol evaluation for the restoration of neural control of both limbs [39,40].

This investigation does have limitations that highlight potential areas for future investigation. The small sample size limits the generalizability of these data, but the two groups were tightly matched on age, biological sex, activity level, and year in school to limit confounds. This was a secondary analysis from a study that primarily investigated between group differences in neural activity for involved limb movement, and therefore an a priori power analysis was not conducted on the current sample. Thus, there is a risk of committing a type I statistical error (finding a significant difference when indeed there is none) for between-group comparisons. Additionally, although an odd number of subjects, our sample was close to a 50/50 split of male to females enrolled. However, during lower extremity movement, prior work demonstrates that healthy males and females may differentially activate their cortex for lower-extremity motor control [41]. Future work should investigate sex-specific neuroplasticity after ACLR. The greater MFG activity could be secondary to other neural compensations related to sensorimotor processing since our fMRI task was not designed to challenge cognitive processing. Previous literature demonstrates that after ACLR, for involved limb movement, there is greater functional connectivity between the frontal cortex (cognition) and parietal and occipital regions responsible for processing multiple sensory stimuli [15]. Therefore, for uninvolved-limb motor control, the MFG might also be a key contributor for cognitive–sensory connectivity. Additionally, to examine the temporal role of the MFG in motor control after ACLR, other neurophysiological measures such as electroencephalography (EEG) with and without dual-task should be used [18]. Furthermore, since no previous study has aimed to evaluate neural activity associated with uninvolved limb movement, our study could be used to power future research investigating sensorimotor control of the healthy limb. Future research should also continue to explore pre-injury and post-ACLR clinical metrics associated with the underlying neurophysiological alterations in knee movement in attempt to improve clinical feasibility of assessment and treatment techniques after ACLR [20,42,43]. For example, after two weeks of eccentric cross-training, patients with ACLR demonstrated improved spinal reflex excitability, cortical excitability, and a reduction in frontal cortex brain activity [44].

## 5. Conclusions

The findings of this preliminary investigation implicate that during a simple uninvolved knee movement, ACLR participants demonstrate greater neural activity within the MFG relative to healthy matched controls. Greater neural activity after ACLR potentially implies greater utilization of cognitive resources for simple knee motor control and movement planning for the uninvolved knee. Our findings indicate that the uninvolved limb should not serve as a neurologic reference or control extremity and that neural activity might be a contributor to uninvolved-limb functional neuromuscular deficits.

## Figures and Tables

**Figure 1 brainsci-15-00109-f001:**
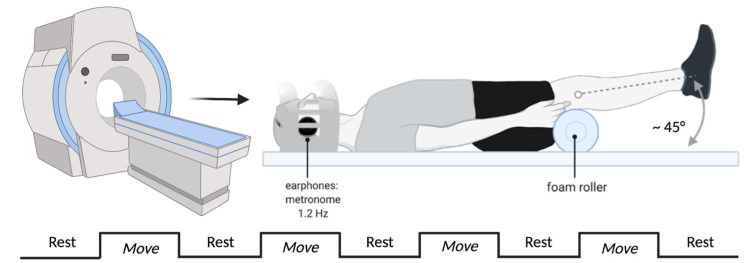
Experimental fMRI set up for knee flexion and extension task paced by an auditory metronome through approximately 45-degrees of motion. The image was created with BioRender.com.

**Figure 2 brainsci-15-00109-f002:**
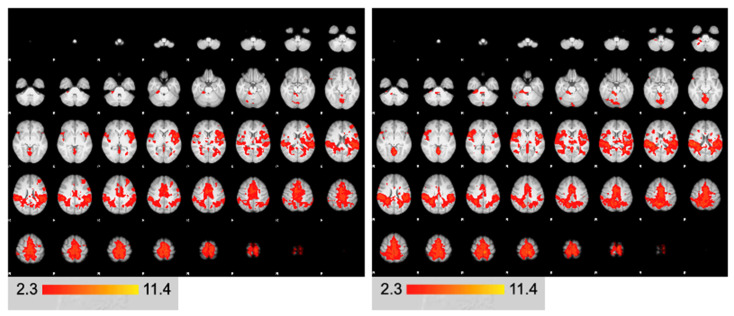
Group average activation maps (**Left**: ACLR; **Right**: Control).

**Figure 3 brainsci-15-00109-f003:**
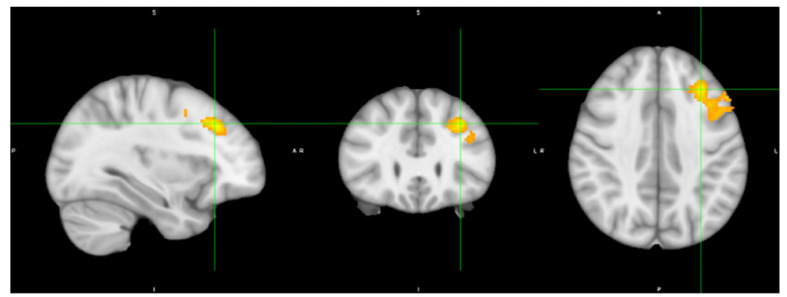
ACLR increased task-based neural activity within the middle frontal gyrus/frontal pole during uninvolved limb movement.

**Table 1 brainsci-15-00109-t001:** Means ± standard deviations for group demographics.

	ACLR (mean ± SD)	Control (mean ± SD)	Significance (α=0.05)
**Gender (male/female)**	8F/7M	8F/7M	
**Age (years)**	21.53 ± 2.7	23.33 ± 2.7	*p* = 0.081
**Height (cm)**	173.22 ± 10.0	174.92 ± 9.7	*p* = 0.643
**Weight (kg)**	72.15 ± 16.1	72.14 ± 15.4	*p* = 0.998
**Tegner Activity Level**	7.40 ± 1.1	7.33 ± 1.0	*p* = 0.863
**Time Since Surgery (months)**	43.33 ± 33.1	__	__
**Graft (Patella Tendon/Hamstring)**	2 PT/13 HS	__	__

Abbreviation: ACLR, anterior cruciate ligament reconstruction.

**Table 2 brainsci-15-00109-t002:** Neural-activity-associated uninvolved knee flexion-extension task in ACLR group.

Cluster Index	Brain Regions	Voxel	*p*-Value	Peak MNI Voxel			Z Stat-Max
				x	y	z	
1	Middle Frontal Gyrus/Frontal Pole	843	0.00184	−32	28	38	3.81

Abbreviation: MNI, Montreal Neurological Institute and Hospital coordinate system.

## Data Availability

The data presented in this study are available on request from the corresponding author due to the nature of neuroimaging datasets being very large and requiring specialized software to handle. The raw analyses output data report is provided.

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
