# Peer review of "Neural Activity for Uninvolved Knee Motor Control After ACL Reconstruction Differs from Healthy Controls"

_brainsci, 2025, doi:10.3390/brainsci15020109_

Round 1

Reviewer 1 Report

Comments and Suggestions for Authors

Neural Activity for Uninvolved Knee Motor Control after ACL Reconstruction Differs from Healthy Controls

This study aimed to assess differences in neural activity using fMRI during uninvolved limb motor control after anterior ligament reconstruction (ACLR) in 15 participants (8 female) as compared to 15 matched controls. A cyclic knee-extension task paced to 1.2 Hz was used. Results highlight a greater neural activity only for one brain region

This article is very interesting and tackles a little-treated subject: the central nervous system (CNS) adaptations contributing to reinjury risk to the involved side. 

At the end, the introduction should try to be more specific by drawing on other work that discusses motor control during unilateral vs. bilateral muscle contractions and the associated neurophysiological mechanisms. 

The methodology appears to be rigorous but is not sufficiently detailed/reported on 2 aspects: performance of the motor task and measurements of neural activity. 

Results section should clearly report all findings of interest with regards to the goal (targeted brain regions etc.).

Overall, this article is short and could benefit from further information. See specific comments below for your convenience.

Abstract:

Line 34. This is truncated after “The ACLR group demonstrate…”

Introduction

The scientific background of the present study is well introduced and straightforward.

One possible lack is information on neural activity from fMRI (or other brain techniques such as EEG or fNIRS) in both involved and uninvolved knees during movement. In other terms basic knowledges when working with unilateral muscles and some changes after de/training: Unilateral strength training preserves strength and muscle cross-sectional area in an opposite immobilized limb for instance.

Lines 67-69, it is stated: “of neural activity occurring in brain regions responsible for  cognitive processing, visual-spatial memory, cross-modal sensory integration, and descending motor control.” And Line 70-71 “as bilateral spinal and supraspinal 70 alterations in quadriceps neuromuscular responses occur”. 

This raises the question of which phenomenon (e.g. cross-training) can promote bilateral supra/spinal changes and which cortical and possibly subcortical brain regions are targeted (i.e. motor areas?). Could the authors expand a bit and provide some key points before proposing a rather large hypothesis "alterations in neural activity"?

Methods

Line 102: motor task

How were participants familiarized with the flexion/extension task to produce smooth movements? 

- Did you measure the kinematics of the movement? If not, how was the movement quantified and controlled? visual/manual counting... Please specify.

-What metrics did the authors use to quantify motor performance during each 30-s task?

-What were the instructions given to the participants during the block paradigm? the positioning of the arms and the overall attitude in order to relax the muscles and therefore only contract the muscles targeted by the knee flexion/extension movement.

L112. fMRI and brain areas.

It is rather strange that no information was indicated on the brain regions targeted (see previous comments for the introduction) when analyzing the brain (cortical?) activity. Please amend accordingly.

L100. Can did the authors provide a BOLD signal for a block of repletion in one subject and give a clear pipeline of analysis of these signals to get “neural activity” as suggested by the authors. 

Results

L125. Only the left middle frontal gyrus/frontal pole brain regions (not reported in the methods) were indicated with a figure and a table. Please clarify (from the beginning) which brain regions were indicated. Authors appears to report only significant findings. You should also report all other non-significant results, often more important to consider. Table 2 is incomplete (only one cluster indicated) accordingly. Please revise it.

Discussion

As the previous sections are not fully informed, some parts of the discussion (especially the first section) need to be revised when comparing the literature.

L201-202: Please justify. Why should functional connectivity analysis be used here? More research quantifying neural activity for some brain regions during flexion/extension movement is needed first.

Any conclusion has been suggested.

Author Response

Responded to all comments in the attached file

Reviewer 2 Report

Comments and Suggestions for Authors

First of all congratulation on submitting your paper and thank you for the invitation to review your interesting study.  The following suggestion could improve the quality of your research.

INTRODUCTION

-          Please delineate more precisely the specific research gap that the study aims to fill, especially in the context of existing study concerning the evaluation of Neural Activity after ACL reconstruction.

-          The introduction would benefit from a clearer statement of the study's objectives and hypotheses.

METHOD

-          The manuscript does not clearly justify the chosen sample size or discuss how it ensures sufficient power to detect expected results.

-          Why the date of approved institutional review board is 2012?

-          The information concerning institutional review board in Line 81 are repeated in line 93

-          Inclusion/exclusion criteria should be better explain, moreover for healthy control group

RESULT

-          The font of tables are different from the font of the text.

-          Statistical analysis paragraph concerning the test utilized should be added

DISCUSSION

-          The most limitation of this study is the impossibility to stratify female from male. Infect female differ from male after ACL Reconstruction, it seems that female suffer more than male concerning altered brain activations

*Mancino F, Gabr A, Plastow R, Haddad FS. Anterior cruciate ligament injuries in female athletes. Bone Joint J. 2023 Oct 1;105-B(10):1033-1037. doi: 10.1302/0301-620X.105B10.BJJ-2023-0881.R1. PMID: 37777208.

-          An other limitation is the sample size. Including in our limitation the small sample size would provide a more balanced view

-          The suggestions for future research should be improved, including the exploration of other outcomes measure, such as transcranial magnetic stimulation or EEG and relative physiotherapy treatments that could improve the altered brain activations observed. For example dual task protocol, in which active exercise is associated with concomitant cognitive traing represents represents an emerging new modality for many disorders that present both physical and cognitive symptoms. Please take in to consideration the following article concerning the use of dual task protocol in rehabilitation:

*Deodato M, Granato A, Buoite Stella A, Martini M, Marchetti E, Lise I, Galmonte A, Murena L, Manganotti P. Efficacy of a dual task protocol on neurophysiological and clinical outcomes in migraine: a randomized control trial. Neurol Sci. 2024 Aug;45(8):4015-4026. doi: 10.1007/s10072-024-07611-8. Epub 2024 May 29. PMID: 38806882; PMCID: PMC11255006.

*Wang L, Yu G, Chen Y. Effects of dual-task training on chronic ankle instability: a systematic review and meta-analysis. BMC Musculoskelet Disord. 2023 Oct 13;24(1):814. doi: 10.1186/s12891-023-06944-3. PMID: 37833685; PMCID: PMC10571247.

Author Response

(The authors gave the same response as above.)

Round 2

Reviewer 1 Report

Comments and Suggestions for Authors

In the revised version, the authors have addressed all previous concerns. This pilot study provides the first evidence of whole-brain adaptations to uninvolved knee motion in patients with a history of ACLR. 

Reviewer 2 Report

Comments and Suggestions for Authors

I thank authors for the work done